# Effect of a Virtual Reality-Based Restorative Environment on the Emotional and Cognitive Recovery of Individuals with Mild-to-Moderate Anxiety and Depression

**DOI:** 10.3390/ijerph18179053

**Published:** 2021-08-27

**Authors:** Hongqidi Li, Wenyi Dong, Zhimeng Wang, Nuo Chen, Jianping Wu, Guangxin Wang, Ting Jiang

**Affiliations:** 1Department of Psychology, School of Humanities and Social Sciences, Beijing Forestry University, Beijing 100083, China; lihongqidi@bjfu.edu.cn (H.L.); dwypsyche@outlook.com (W.D.); af_chennuo@bjfu.edu.cn (N.C.); wangguangxin2016@bjfu.edu.cn (G.W.); 2Faculty of Psychology, Beijing Normal University, Beijing 100875, China; 202028061042@mail.bnu.edu.cn (Z.W.); psytingjiang@bnu.edu.cn (T.J.)

**Keywords:** restorative environment, virtual reality, presence, anxiety and depression, cognitive recovery

## Abstract

In this study, restorative environment theory and virtual reality (VR) technology were combined to build different 3D dynamic VR interactive scenes. We discuss the effects of a VR restorative environment on the emotional and cognitive recovery of individuals with mild-to-moderate anxiety and depression. First, we built a VR restorative garden scene, divided into four areas: forest, lawn, horticultural planting, and water features. The scene was verified to have a good recovery effect in 26 participants. Then, 195 participants with mild-to-moderate anxiety and depression were selected as experimental subjects. Through psychological testing and EMG (Electromyography) and EEG (Electroencephalography) data feedback, we further explored the differences in the sense of presence in VR restorative scenes and their effect on individual emotional and cognitive recovery. The results showed that (1) both the restorative environment images and the VR scenes had a healing effect (the reduction in negative emotions and the recovery of positive emotions and cognition), with no difference in the subjective feeling of recovery among the different scenes, but the recovery score of the VR urban environment was higher than that of the natural environment (differing from the results in real environments); (2) a high sense of presence can be experienced in different VR scenes, and interactive activities in VR scenes can provide a great presence experience; (3) the recovery effects of VR restorative environment on emotion and self-efficacy are realized through the presence of VR scenes; (4) a VR restorative environment is helpful for the emotional improvement and cognitive recovery of individuals with mild-to-moderate anxiety and depression. VR urban scenes also have good recovery effects. In terms of cognitive recovery, self-efficacy improved significantly. In addition, from the perspective of EEG indicators, the VR restorative scene experience activated the prefrontal lobe, which is conducive to cognitive recovery in individuals with mild-to-moderate anxiety and depression. In terms of emotional improvement, negative emotions were significantly reduced in the different VR scene groups. In conclusion, we further explored ways to help individuals with mild-to-moderate anxiety and depression, in order to promote the development and application of mental health.

## 1. Introduction

With rapid urbanization, the attention of individuals is consumed and pressure increases are experienced due to a high intensity of study and work, continuously affecting their physical and mental health [1,2]. This has led to anxiety, mood disorders, and even schizophrenia and other psychological and psychiatric problems becoming more common [3]. Exploring the restorative power of strategies focused on stress states, such as anxiety and depression, has become a topic of concern for researchers both at home and abroad [4,5]. Restorative environments (e.g., natural environments) can restore attention and reduce stress, effectively improving physical and mental health [6,7,8,9], thus providing the possibility of meeting the physical and mental needs of the public. In addition, virtual reality (VR) technology has shown great potential in the field of mental health in recent years [10], and the manner in which to combine restorative environments and VR technology to regulate the mental health of the general public has become an important topic. Therefore, it is of theoretical value and practical significance to explore the realization of restorative environments and their healing effect in VR.

### 1.1. Restorative Environments and Their Restorability

The restorative environment concept was first put forward by environmental psychologists [11], which refers to a kind of environmental setting that, to a certain extent, can replenish, restore, or update tired or wasted psychological resources; that is, a restorative environment is an environmental setting that has the effect of restoring and updating those physical and mental resources and abilities that are constantly consumed under urbanized environments. A restorative environment should have four characteristics: being away, extent, fascination, and compatibility [12]. Being away requires the environmental setting to make individuals feel far away from their daily environment and disturbances, as well as avoiding the use of directed attention. Extent refers to the richness and continuity of scenes, which pleases the mind and promotes exploration. Fascination requires that contextual information be noticed without effort. Environmental compatibility matches one’s goals and inclinations. Visual appeal, naturalness, and lack of human habitation are three important elements of a restorative environment [13].

The theoretical basis of the research on the restorative environment includes attention restoration theory (ART) proposed by Kaplan and stress reduction theory (SRT) proposed by Ulrich, also known as psychological evolution. ART points out that a task that requires psychological resources will arouse directional attention. If the duration and intensity of the task reaches a certain degree, even if the target is pleasant, it can also cause mental exhaustion. Natural scenes can provide active involuntary attention to stimulation, causing the attentional system to resume. The theory explains how to attract involuntary attention to the natural environment and how to recover cognitive ability. SRT states that, when individuals under stress are exposed to medium complexity, visual focus, depth, mystery, no potential threat, uniform ground texture, deflecting visual field, and environments containing plants and water, their attention will be attracted, thus blocking negative thoughts, and so, their emotions can change from negative to positive. The disturbed physiology returns to a balance, and so does cognitive behavior. These two theories have different focuses in terms of concept and mechanism. The former focuses on fatigue in order to explain the recovery of individual attention resources and the change in cognition during natural experiences, while the latter focuses on the individual emotional response to nature. In recent years, the two theories have shown a shift in trend from opposition to integration [14]. 

Hartig elaborated on restoration, which refers to regaining the physical, psychological, and social abilities that are lost in the process of adapting to the external environment [15]. Recovering environmental perception includes physiological, emotional, and attention aspects. The observable recovery process includes positive emotional change, a decline in the autonomous arousal level, improvement in the completion of directed attentional tasks, and so on [16,17]. Experiences in natural environments lead to feelings of healing [18,19]. Restorative environments help to improve cognitive function, improve the emotional state, and reduce stress [20,21,22,23,24,25,26,27]. In addition, the recovery effect is also reflected in cognitive recovery [28]: it has been found that natural connections are associated with cognitive style [29], and the restorative environment experience can restore and stimulate cognitive and thinking creativity, allowing individuals to achieve a better mental state and work performance [30,31]. These restorative effects have been fully proven when using real natural scenes [27,32], pictures and video materials [33,34,35], and immersive VR [33,36,37,38]. Related EEG (Electroencephalography) studies have promoted our understanding of ART. Natural restorative experience is related to α and θ bands, and α-θ activity may play an important role in attentional recovery [39]. During early cognition, restorative natural scenery caused more alpha oscillations [40]. The enhancement of α waves indicates possible sensory inhibition, supporting the hypothesis that restorative nature may occupy less attention and cognitive resources, and that the natural environment contributes to cognitive recovery [41].

Research on rehabilitation of anxiety and depression in virtual reality is very common. Presenting virtual fear stimuli in exposure therapy to treat anxiety, using immersive games to distract patients to relieve their pain, or presenting a natural environment to achieve relaxation [42,43,44] are the most widely used virtual reality scenarios in psychology. These types of virtual reality scenes have not received attention by researchers in meta-analysis so far [45,46]. Restorative environments have multiple effects of relieving anxiety, pain, and stress. Restorative virtual reality scenes are mostly the natural environment, including natural scenes [47,48] and virtual scenes [43,49]. We used the restorative landscape elements and designed interactive activities (such as fishing, water plants), so participants can not only walk freely to enjoy the environment, but also experience the fun of playing inside it and, thus, can enhance the restorative experience.

The measurement of environmental resilience includes subjective reporting and objective measurement. The Perceived Restorative Scale (PRS) covers the emotional, physiological, cognitive, and behavioral dimensions [50]. The Chinese version of the PRS has been widely used in the study of restorative environments [51]. The restorative effect can be observed and measured by attention and emotion, as well as through physiological indices such as EEG [48,52], fMRI (Functional Magnetic Resonance Imaging) [53], EMG (Electromyography) [54], skin electricity, and heart rate [55]. Research on the restorative effects of the environment has always been a hot topic in the field of environmental psychology, which has theoretical value and practical significance.

### 1.2. The Psychological Application of Virtual Reality (VR) Technology

With the arrival of the intelligent era, new technologies represented by VR have gradually come into public view and have improved the quality of human life [56]. VR technology, based on its high immersion, control, security, and other characteristics, has been widely used, including in psychological research and clinical psychology [57,58,59,60,61,62,63,64,65]. It also plays an important role in the education and training of young people and professionals [66,67]. In a simulated environment, the weather, lighting, and other factors can be controlled, which cannot be controlled in a real environment. Multiple scenes can also be presented in a short time and randomly presented in a sequence, thus breaking the restrictions of time and space and being conducive to multi-sensory experiments and the collection of physiological data [68]. VR technology has also been applied in mental health education [69]; the prevention, assessment, and intervention of mental illness [70,71,72]; in the medical field [73,74]. Existing studies have confirmed that VR technology can successfully treat delusions, hallucinations, and other psychiatric symptoms, as well as improving the effectiveness and generality of cognitive and social skills, with the recognition that immersive VR therapy is safe, acceptable, and has a long-term effect [75].

In previous studies, the virtual reality scene was in the form of video materials [47,76], 360° image composited video [77], or artificial scenes created by technology. The VR environment created by technology was generally perceived as restorative as the physical nature environments, and more fascinating and coherent [49]. Relevant studies have proved that using virtual scenes for education can improve people’s emotions and consciousness [78,79]. The current virtual scene construction technology is riper, and the virtual scene is more flexible and controllable and can also be used to explore the interaction between people and the scene. Therefore, this method of constructing interactive virtual reality scenes is also used in this study.

VR is a type of completely immersive computer simulation experience utilizing a three-dimensional interactive virtual environment. A VR system consists of a computer, a head-mounted display, headphones, and motion-sensing gloves [80]. Compared to television and other media, VR can cause a greater degree of improvement in positive emotions, which seems to be mediated by a stronger sense of presence and natural interaction [76]. Presence, also known as a sense of presence, is a sense of being in an intermediary environment [81], which is a psychological state of being. It is also a specific cognitive process [82]; that is, the psychological experience of individuals feeling themselves in the virtual environment created by computer monitors. There are three main categories: personal, social, and environmental. Personal presence is the extension of the experiencer’s sensory emotions to the virtual environment, resulting in an immersive experience and a sense of psychological participation. Social presence is the degree of coexistence and interaction with other creatures in the virtual environment, while environmental presence is the degree to which experiencers respond to the environment [83]. Self-efficacy is an intermediary between social presence and cognitive presence [84]. Researchers have proposed that it is necessary to further explore the differences in VR technology in different environments and the impact of such differences on individual self-efficacy [85].

Sheridan proposed the development of a scientifically useful presence metric and determined its significance and value in remote operation applications [86]. Mel and Slater believed that presence is not the same as immersion [87]. They established an equation, where the left side contains presence, and the right side contains the factors of immersion and the differences between individuals. Time [88], content [89], user characteristics [90], attention resources and participation, immersion, and interaction [91,92] are all factors that affect the sense of presence. Its measurement methods include objective measurement and subjective reporting. Observing postural response amplitude, measuring heart rate and skin electricity, and observing facial expressions, gestures, head movements, eyes, intonation, and other social behaviors can reflect the sense of presence [93,94,95]. The Presence Questionnaire (PQ) was developed and revised in 2005 based on four dimensions [96]: control, sensation, distraction, and reality. The virtual environment physiologically stimulates the emotional state of the user, which, in turn, enhances their sense of presence [97]. It is regulated by the environment, which attracts the senses and attention and promotes active participation, which may be real, virtual, symbolic, or some combination thereof [98]. Schubert argues that Glenberg’s embodied cognitive framework is the means used to explain the sense of presence. Presence is the result of media perception [99]. In the process of presence perception, a mental model of virtual three-dimensional space is constructed. This model is composed of possible behaviors in space, and the body’s actions are the core. In other words, because the virtual environment is perceived according to specific actions, the mental state of presence is generated. The production process of the sense of presence is consistent with the theory of embodied cognition that cognition is generated through physical experience and its activity mode. Therefore, in this study, EMG is used as another indicator of presence.

### 1.3. A Restorative Environment Based on Virtual Reality, Emotion, and Cognition

The current research on VR restorability mainly focuses on urban and natural scenes with the natural elements and natural analogues and experience of space and place [100]. Studies have pointed out that simulating the experience of an urban or natural environment has a positive effect on mental health [101,102], and virtual nature may be a beneficial supplement to actual nature. It helps people to reconnect with the real natural world and promotes interactions between people and nature [103]. In addition, VR environments have the restorability of physical forest environments and are even more attractive and coherent. Some researchers have applied VR technology to the study of psychological recovery due to exposure to a natural environment [104], proving the stimulating effect of forest environments and natural sounds in VR on the recovery of psychological stress. Exposure to VR-based natural environments can promote stress reduction and the improvement of mood [105], which provides a practical basis for exploring VR-based restorative environments. Although natural environment experiences can bring restorative feelings [18], existing studies have also shown that participants in virtual natural environments show more positive psychological effects, compared to in urban environments [47]. On the one hand, restorative environments, especially environments based on VR, are not necessarily only natural environments, and so, it is necessary to explore different healing factors. On the other hand, immersive VR systems can create intimate interactions with the environment through headgear and controllers, thus creating high-presence experiences from the perspective of a first-person narrative, which is conducive to reducing boredom [76], stimulating positive emotions, and improving the sense of self-efficacy [106]; however, existing studies on the restoration of positive and negative emotions and cognitive functions from the perspective of VR-based environment settings and interactions are few, and further studies on the mind–body action mechanism are needed.

On the basis of previous studies, we explore the common recovery theories (ART and SRT), which have been predominantly tested on the cognitive and emotional recovery of individuals in former studies. Moreover, researchers investigate the positive effect of a restorative environment based on VR. However, how to design restorative virtual scenarios based on recovery theories is unclear and the effectiveness of VR scenarios is of great interest to the interdisciplinary research field [100]. Therefore, this study began by focusing on the general population and gradually focused on individuals with mild-to-moderate anxiety and depression tendencies. Taking an immersive VR-based restorative environment as the research material, from the perspective of presence, we verified the recovery effect on individual emotion and cognitive activity and compared the differences in the recovery effect under different VR restorative environments.

## 2. Study I: Design, Implementation, and Verification of a VR Restorative Environment

### 2.1. Purpose and Hypothesis

In this part, a restorative VR environment is designed and realized, the purpose of this study was to verify its recovery effect, to select the experimental materials for the next step from it. Based on the results of previous study of restorative environment, images and VR scenes of restorative environment have healing effects. The hypotheses were made for Study I:(1)The 2D images of initial restorative scenes used in this study have recovery effect.(2)The VR restorative scenes we used have recovery effect.

### 2.2. Design and Implementation of a VR Restorative Environment

Based on environmental psychology, landscape design, rehabilitation medicine, and other relevant materials, appropriate environmental elements were selected to construct a restorative garden, which included seven environmental zones: forest area, meditation area, flowers area, lawn area, gardening area, rest and interaction area, and water scenic area. The researchers used hand-drawn images and PS (CC2019, Adobe Systems Inc., San Jose, CA, USA), AI (CC2019, Adobe Systems Inc., San Jose, CA, USA ), CAD 2019 (Autodesk, Inc., San Rafael, CA, USA), Sketchup 2019 (Trimble Inc., Sunnyvale, CA, USA), and other design software to draw the scenes and invited experts to evaluate and modify the environmental elements, paths, colors, and so on. After this, Lumion 8.0 software (Act-3D Corp., Greenfield, IN, USA) was used to present the preliminary restorative virtual reality environment. Finally, the interactive development system of Unity was utilized to construct an immersive VR environment, in which users could freely roam within the limited space.

### 2.3. Experiment 1: Recovery Validation of Scenes

#### 2.3.1. Participants

The initial study participants were 10 undergraduates (*n*_male_ = 4) from Beijing Forestry University, with an average age of 18.80 years, who were majoring in applied psychology, law, civil engineering, and so on. After a normality test, data from eight participants were included in the analysis.

#### 2.3.2. Materials and Instruments

A total of 30 images were captured from the different angles of the designed restorative environment in Lumion 8.0, as shown in Figure 1 and Figure 2.

The Restoration Environment Scale (RES) was used to evaluate the restoration effect of the VR scenario. This scale was compiled based on ART theory and is the first Chinese version of the RES, which can be divided into three dimensions: being away, fascination and compatibility, and abundance. The Cronbach’s α coefficients of the total scale and the three sub-scales ranged from 0.769 to 0.936, and the split reliability distribution ranged from 0.695 to 0.903, indicating high reliability and validity.

An Asus PU554U (CPU: Intel^®^ Core^TM^ i7-7500U@2.70 GHz; GPU: NVIDIAGeForce940MX, Asustek Computer Inc., Shanghai, China) computer was used to display the pictures captured from the scenes.

#### 2.3.3. Procedure

After the participants signed the informed consent, they entered the laboratory and watched 30 pictures, focusing on each picture for 5 s. The pictures were presented in random order. After viewing, they filled out the RES.

#### 2.3.4. Results

Statistical analysis using IBM SPSS STATISTICS 23.0 (IBM Corp., Armonk, NY, USA) software showed that the average score (standard deviation) of the recovery environment, with scores of 4.06 (1.00) for being away, 5.29 (1.08) for fascination and compatibility, and 2.72 (1.23) for abundance. The results indicate that the initial restorative scenes had a certain recovery effect, in which the characteristics of being away, attraction, and compatibility were better than the characteristic of abundance; the experimental effect may be affected by the number of participants and fewer viewing materials. In addition, in this experiment, we only presented two-dimensional pictures, such that the experience of the participants may differ from that in VR, meaning further exploration should be carried out using VR equipment. The presentation of scenes should be combined with the technical realization effect, which was suggested to improve the richness of scenes, and to select forest, lawn, horticultural planting, and waterscape as the main environmental partition.

### 2.4. Experiment 2: Verification of Recovery in a VR Environment

#### 2.4.1. Participants

The participants of the second experiment were 16 undergraduate students (*n*_male_ = 9) from Beijing Forestry University, with an average age of 22.31 years, majoring in applied psychology, landscape architecture, computer science, and engineering. After completing the experiment, the participants were paid 20 yuan.

#### 2.4.2. Materials and Instruments

The VR scenes experienced by the participants were in a restorative garden constructed using the Unity interactive development system, including four environmental partitions, i.e., forest, lawn, garden, and waterscape. The VR equipment and host equipment were the same as in Experiment 1, and the RES was used to evaluate the experimental scenes.

#### 2.4.3. Procedure

The experimental process is shown in Figure 3. The participants signed an informed consent form. Then, the experimenter explained how to operate the controller (including forward, backward, left, right, transient, and trigger pulling). The participants put on the VR device and entered the experimental scenes. After completing the adaptation training, the participants had 10 min to roam within the scenes. In the process of experience, the participants freely explored the scene. They had to reach the forest, lawn, garden, and waterscape, and the order in which they were exposed to these elements was random. During the experiment, the experimenter was not far away from the participants in order to ensure their safety. At the end of the experiment, the experimenter asked the participants about their feelings. Finally, the participants filled out the RES.

#### 2.4.4. Results

The IBM SPSS STATISTICS 23.0 software was used for statistical analysis, and the data of the total scale and three sub-scales of the RES were found to be normally distributed. The average score for the restorative environment was 4.11 (0.65), while it was 4.63 (1.07) for being away, 4.75 (1.12) for fascination and compatibility, and 2.96 (1.04) for abundance. Taking the mean of the seven-point scale (4) as the comparison standard, an independent samples *t*-test was conducted for the score of the Restorative Environment Scale. The results showed that there was no significant difference between the total score and the mean of the scale (*t*(15) = 0.696, *p* > 0.05), but the means of three dimension scores (*t*_being away_(15) = 2.335, *p* < 0.05; *t*_attraction and compatibility_(15) = 2.668. *p* < 0.05; *t*_rich_(15) = −3.996, *p* < 0.01) were significantly higher than the average score (4). We found that the VR restorative environment, in terms of being away, fascination and compatibility, and abundance, was better than viewing 2D images.

## 3. Study II: Presence and Recovery Effect of Experiencing a VR Restorative Environment

To further investigate the differences in the presence of VR environments and their interactive activities, and to verify their effects on individual emotional and cognitive recovery, 20 participants were recruited to conduct a pre-experiment, for which five scenes were identified: a watering task in the garden area and fishing task in the waterscape area with better interactive experiences, controller-free and controller garden scenes, and a VR urban environment (which was added as a control group). These five virtual scenes were used as an intervention in individuals with mild-to-moderate anxiety and depression, as well as to discuss the effect of a VR restorative environment, the sense of presence, and its effects on the emotions and cognition of individuals, through scale measurements and relevant physiological data.

### 3.1. Purpose and Hypothesis

The purpose of this study was to explore the intervention effect of the presence of a VR restorative environment on individuals with mild-to-moderate anxiety and depression from the perspective of emotion and cognition. Based on the results of Study I and the pre-experiment, further hypotheses were made for Study II:(1)There will be differences in subjective restoration and the sense of presence of different VR restorative scenes. The subjective restoration of VR restorative scenes will be higher than that of VR urban scene. The presence of VR restorative environment with interaction will be better than that in other intervention groups.(2)Different VR restorative environment experiences will have different healing effects on the degree of change in individual emotions and self-efficacy for individuals with mild-to-moderate anxiety and depression.(3)VR restorative scenes (Env2~Env5) will contribute to improve positive emotions, reduce negative emotions, and improve self-efficacy of individuals with mild-to-moderate anxiety and depression. The VR urban scene (Env1) will have the opposite effect to VR restorative scenes. Different VR restorative environment experiences will have different healing effects on the directions of change in individual emotions and self-efficacy.(4)The recovery impact of the VR rehabilitative environment on people was probably realized through the presence of VR scenes.(5)The differences of presence will also be reflected in EMG: compared to the baseline, indicators of physical participation (contraction of the brachioradialis muscle of the participants’ arm) in the VR scene experience will improve.(6)VR restorative scenes will be conductive to the cognitive recovery of individuals with mild-to-moderate anxiety and depression, which will be reflected in EEG indicators: prefrontal alertness and engagement will be increased, and the calming signal index will be decreased in the VR restorative environment experience.

### 3.2. Methods

#### 3.2.1. Participants

A total of 369 undergraduate and graduate students from Beijing Forestry University and Beijing Normal University were recruited, with normal vision or corrected vision, no major physical or mental diseases, and good attention and executive ability. The screening criteria were state anxiety and depression scale scores ≥40 and ≤60. Among them, 195 were eligible to participate in the experiment, and 189 valid data were obtained. A total of 189 participants (*n*_male_ = 76; *n*_female_ = 113) were eligible for the experiment, with an average age of 20.26 years (2.58).

The participants were divided into a VR urban environment visual experiencing group (Env1, *n*_1_ = 39), a VR restorative environment visual experiencing group (Env2, *n*_2_ = 35), a VR restorative environment interactive experiencing group (Env3, *n*_3_ = 37), a VR restorative environment with fishing interaction group (Env4, *n*_4_ = 40), and a VR restorative environment with watering interaction group (Env, *n*_5_ = 38). The participants earned 10 yuan after completing the experiment.

#### 3.2.2. Experimental Design

A completely randomized experimental design was used. Each participant was randomly assigned to one of the experimental groups, including the VR urban environment visual experiencing (Env1), VR restorative environment visual experiencing (Env2), VR restorative environment interactive experiencing (Env3), VR restorative environment with fishing interaction (Env4), and VR restorative environment with watering interaction (Env5) groups, for a total of five experimental groups. This study was carried out in October 2020, following approval from the Ethics Committee of the Department of Psychology, College of Humanities and Social Sciences, Beijing Forestry University.

#### 3.2.3. Materials and Instruments

The experimental materials and instruments were as follows:

VR scenes constructed using the Unity interactive development system—urban and restorative environments (four areas, including lawn, garden, water feature, and forest, as well as a visual experience, an interactive experience, and two interactive activities, the difficulty of the interactive activities is moderate and easy to learn, which will not burden the participants). Screenshots of the scenes are shown in Figure 4, Figure 5, Figure 6, Figure 7, Figure 8 and Figure 9.

VR equipment: An HTC Vive Pro Eye set (HTC Corp., Hsinchu, Taiwan, China), including one helmet, two handles, two 2.0 locators, two locator brackets, and a data cable.

Computer equipment: Alienware (CPU: Intel^®^ Core^TM^ i7-8700k@3.70 GHz; GPU: NVIDIA GeForce GTX 1080 Ti, Dell, Inc., Round Rock, Texas, USA).

A set of BIOPAC systems (BIOPAC Systems, Inc., Goleta, CA, US), Inc MP160 physiological permeameters: Used to detect and record the electrical activity during the process of the intervention. An EMG100C amplifier was required for EMG signal acquisition, along with two LEAD110S shielded conductors, one LEAD100A unshielded conductor, and three disposable patch electrodes. An EEG100C amplifier was required for EEG signal acquisition, along with two Lead110 shielded leads, one Lead100 unshielded lead, and three disposable patch electrodes.

The Restorative Environmental Scale (RES): the same as used in Study I.

The Presence Questionnaire: Composed of 29 items, including involvement, sensory authenticity, adaptation, and interface quality [98]. It has good reliability and validity.

State–Trait Anxiety Inventory (STAI): The anxiety of participants was assessed with a 40-item scale, on a scale of 1–4: questions 1–20 comprised the State Anxiety Inventory (STAI, Form Y-I, S-AI), while questions 21–40 comprise the Trait Anxiety Inventory (STAI, Form Y-II, T-AI); the former describes short-term unpleasant emotional experiences (e.g., tension, fear, and anxiety), often accompanied by vegetative nervous system function, while the latter is typically used to describe the relatively stable anxiety tendency as a personality characteristic with individual differences [107].

The Self-Rating Depression Scale (SDS): Contains 20 items and is rated from 1 to 4. It is easy to use and can fairly intuitively reflect the symptoms, severity, and changes in the depression state [108].

The Positive and Negative Affect Scale (PANAS): Used to assess the positive and negative emotions of participants. Chinese scholars have conducted a study on the applicability of PANAS among the Chinese population [109] and introduced an effective tool for the assessment of two emotional dimensions, which demonstrated the good reliability and validity of the scale.

General Self-Efficacy Scale (GSES) (Chinese version): Used to evaluate self-efficacy, including 10 items. The Chinese version has good cross-cultural applicability, reliability, and validity [110].

#### 3.2.4. Procedure

The experimental procedure is shown in Figure 10.

After understanding the experimental content and signing an informed consent form, the participants began the experiment, filled out the Positive and Negative Affect Scale and the General Self-Efficacy Scale, and were then equipped with the MP160 physiological equipment. First, we wiped the skin surface of the participants with alcohol and normal saline, then pasted on the disposable patch electrode. For EMG measurements, positive and negative EMG input signals were placed on the right wrist, the flexor carpi radialis muscle, and a GND (Ground) electrode was placed on the elbow to measure the brachioradialis muscle EMG signals. For EEG measurements, positive and negative input signals were placed on the left and right sides of the forehead. A GND electrode was placed behind the ear, at the mastoid of the temporal bone. After setting the original channel and collection parameters, we started collecting EMG and EEG signals. After wearing the physiological equipment, participants were required to sit on a swivel chair and conduct breathing relaxation training to help them stay at rest and relaxed. Baseline physiological data collection was conducted, which was suspended after 3–5 min. Then, the participants wore the VR equipment and adjusted the tightness of the helmet to ensure that their vision was clear, after explaining the required operations in the virtual scene (only for the interactive activity group) to the participants (as shown in Figure 11 and Figure 12).

VR adaptation training was conducted in order to ensure that the individuals could use the handle smoothly and without vertigo. Successful operation was defined as the successful completion of the instructions “forward, back, direction adjustment, and pull the trigger” after entering the VR environment. After this, the VR experience officially started, and the scenes and interactive activities were experienced in the VR environment following the instructions of the main test. The experience lasted for 10 min.

After the experience, the VR devices, physiological instruments, and disposable patch electrodes were retrieved from the participants. If the participants were in good condition, they filled out the PANAS and GSES, as well as the RES and PQ. If the participants experienced cyber sickness, they could quit the experiment at any time.

### 3.3. Results

#### 3.3.1. Differences in the Environmental Recovery of Different Scene Experiences

SPSS 23 was used to conduct statistical analysis on the restorative environmental data. The scores of RES were in line with a normal distribution or approximately normally distributed. The descriptive statistical results are shown in Figure 13.

One-way ANOVA was used to compare the differences in the scores of the participants during the recovery environment under different activity conditions, consistent with the hypothesis of homogeneity of variance. The ANOVA results were *F* (4141) = 1.594 and *p* = 0.179. There were no inter-group differences in the scores of the recovery environment under different activity conditions, which indicates that there were no significant differences in the subjective feelings of recovery in the participants across the five experimental scenarios. There were no significant differences in the environmental recovery of the VR scenes or their dimensions; however, from the perspective of the descriptive statistical results, the environmental recovery of the VR urban environment visual experience group scored the highest. In terms of being away, attraction, and compatibility, the VR restorative environment visual experience group was the best, while, in terms of the richness dimension, the VR healing environment interactive tour group was the best.

#### 3.3.2. Differences in the Presence Experienced in Different Scenes

The presence questionnaire data were tested for normality. The absolute values of z-score of kurtosis and skewness were all less than 1.96, and the data were found to be normally distributed. The presence scores for each scene are shown in Figure 14.

One-way ANOVA was adopted. The homogeneity of variance test results showed that the sense of presence of each experimental group was in line with the hypothesis of homogeneity of variance. The one-way ANOVA results for the sense of presence of the five groups were *F* (4161) = 5.036 and *p* = 0.001, and there were significant differences in the sense of presence for the different scenes.

The scores of each group’s experienced scenes were LSD-tested, as shown in Figure 14. There were significant differences between Env1 and Env3 (*t*(1) = 3.604, *p* < 0.001), Env2 and Env3 (*t*(1) = 3.648, *p* < 0.001), Env3 and Env4 (*t*(1) = −3.218, *p* < 0.01), and Env3 and Env5 *(t*(1) = −3.239, *p* < 0.01).

#### 3.3.3. Mediating Effect of Presence

PROCESS was adopted to analyze the mediating effect, and Model 4 was selected as the mediating effect model.

The results were shown in Table 1. It showed that the sense of presence played a mediating effect in the effect of RES on positive emotions, β = 0.0441, CI [0.0169, 0.0748], 0 was not included in the Boot confidence interval of the mediating effect test, indicating that the mediating effect was significant, and the mediating effect of presence accounted for 42.20% of the total effect. The sense of presence also had mediating effects on negative emotions, β = −0.0449, CI [-0.0881, −0.0108], 0 was included in the Boot confidence interval of the mediating effect test, indicating that the mediating effect was significant and the mediating effect of sense of presence accounted for 58.01% of the total effect, and it was a negative effect. In addition, the sense of presence played a mediating effect in the effect of RES on self-efficacy, β = 0.0277, CI [0.0004, 0.0611], 0 was included in the Boot confidence interval of the mediating effect test, indicating that the mediation effect was significant, and the mediation effect of presence accounted for 57.11% of the total effect.

#### 3.3.4. Effect of VR Restorative Environment Experience on Emotion and Self-Efficacy

For data analysis, we excluded outliers (the extreme values beyond three standard deviations) from the positive and negative emotional and self-efficacy scores of the five experimental groups before and after experiencing VR scenes. The skewness and kurtosis of the data of positive and negative emotions and self-efficacy were examined. The original data did not conform to the normal distribution, so logarithmic transformation was performed on the data to ensure that the data were in line with a normal distribution or were approximately normally distributed. Figure 15 shows the descriptive statistical results of positive and negative emotions and self-efficacy.

Repeated measurement ANOVA for positive emotion, negative emotion, and self-efficacy was used, respectively, to compare the differences in the scores of the participants during the restorative environment under different groups. For positive emotions, there was significant difference between pre- and post-test, *F* (1141) = 6.984 and *p* = 0.009. The main effect of experience scene was not significant, *F* (4141) = 0.560 and *p* = 0.692. The interaction was not significant, *F* (4141) = 1.562 and *p* = 0.188. For negative emotions, there was significant difference between pre- and post-test, *F* (1141) = 63.215 and *p* < 0.001. The main effect of experience scene was not significant, *F* (4141) = 0.758 and *p* = 0.554. The interaction was not significant, *F* (4141) = 0.391 and *p* = 0.814. For self-efficacy, there was significant difference between pre- and post-test, *F* (1141) = 23.593 and *p* < 0.001. The main effect of experience scene was not significant, *F* (4141) = 0.359 and *p* = 0.837. The interaction was not significant, *F* (4141) = 0.899 and *p* = 0.466. These results indicated that VR scene intervention does have an impact on participants, and the difference between different scenes was not significant.

In order to further clarify how the emotional and self-efficacy score of the participants changed after the scene experience, whether it was significantly increased or decreased, and whether there were different directions of change between different scenes, a paired samples *t*-test was conducted for the positive and negative emotional and self-efficacy scores of the five groups before and after experiencing the VR scenes. The results are shown in Table 2. In the Env1, for positive emotions, there was no significant difference before and after the VR scene experience. For negative emotions, it was significantly reduced after the scene experience. For general self-efficacy, the margin of difference before and after the VR scene experience was significant, self-efficacy increased after the scene experience. It was found that the VR urban environment visual experience significantly reduced negative emotions, but had no significant impact on individual positive emotions or self-efficacy, which was not consistent with our expectations. In the Env2 condition, for positive emotions, there was no significant difference before and after the VR scene experience. For negative emotions, the score of negative emotion after scene experience was significantly lower than before. For general self-efficacy, there was a significant improvement after the VR scene experience. It was shown that the visual experience in the VR restorative environment significantly reduced negative emotions and significantly increased self-efficacy, but had no significant influence on individual positive emotions. In the Env3 condition, for positive and negative emotions, there was a significant decrease after the VR scene experience. For general self-efficacy, there was no significant difference before and after the VR scene experience. It was found that the interactive experience in the VR restorative environment significantly reduced positive and negative emotions, but had no significant influence on individual self-efficacy. In the Env4 condition, for positive emotions, there was no significant difference before and after the VR scene experience. For negative emotions, it was significantly reduced after the scene experience. For general self-efficacy, there was a significant improvement after the VR scene experience. It was found that the VR restorative environment interactive experience with fishing significantly reduced negative emotions and enhanced self-efficacy, but had no significant influence on positive emotions. In the Env5 condition, for positive emotions, there was no significant difference before and after the VR scene experience. For negative emotions, there was a significant decrease after the VR scene experience. For general self-efficacy, it was significantly increased after the scene experience. It was found that the VR restorative environment interactive experience with watering activity significantly reduced negative emotions and enhanced self-efficacy, but had no significant influence on positive emotions.

All of the five virtual scenes significantly reduced negative emotions. The VR restorative environment interactive experience significantly decreased positive emotions. The VR restorative environmental visual experience and VR restorative interactive fishing and watering activity experiences significantly improved self-efficacy. We confirmed the hypothesis that different VR restorative environment experiences have different healing effects on the directions of change in individual emotions and self-efficacy. Among them, the VR urban environment visual experience significantly reduced negative emotions, but did not have a significant impact on individual positive emotions and self-efficacy, which was inconsistent with our expectations.

#### 3.3.5. EMG and EEG Feedback from Different Scene Experiences

The collected EMG and EEG signals were processed digitally using AcqKnowledge 5.0 software (BIOPAC Systems, Inc., Goleta, CA, USA). First, a comb filter was used to set the fundamental frequency of 50 Hz, an IIR recursive filter was used for preliminary filtering, and low-to-high-pass EEG filtering was set in the range of 1–40 Hz. The low-to-high-pass EMG filtering was set in the range of 1–500 Hz, and the EMG and EEG signals after initial noise reduction were obtained. Furthermore, the noise caused by errors in the experiment was manually deleted. Meanwhile, MATLAB 2019A software was used for offline denoising and analysis of the EMG and EEG signals, and the time and frequency domain characteristics of the EMG signals, as well as the average power and ratio of the power spectral density of the EEG signals, were obtained.

The pre-processed EMG signals were extracted for features. The time domain eigenvalues were the root mean square (RMS) and the mean absolute value (MAV), and the frequency domain eigenvalues were the mean power frequency (MPF) and the median frequency (MF), reflecting contraction of the brachioradialis muscle of the participants’ arm, which was used as a body participation index reflecting the VR scene experience. Partly due to the relatively small sample size and the normal test of experimental data showing that they did not follow a normal distribution, the scene before the experience and the experience in the process of electromyographic signal characteristics were separately subjected to the Wilcoxon signed rank test, and we obtained the results: RMS, Z = 7.818 (*p* < 0.001), MAV, Z = 7.818 (*p* < 0.001), MPF, Z = 4.800 (*p* < 0.001), and MF, Z = 2.698 (*p* < 0.001). The significant differences between the time and frequency domain eigenvalues of EMG in the five groups before and during the scene experience are shown in Figure 16. Combined with the descriptive statistical results, the EMG eigenvalue was significantly increased compared to the baseline level during the scene experience, indicating that the scene experience enhanced the contractive activities of the arm muscles, reflecting an increase in body involvement.

The EMG time and frequency domain characteristics in the different experience scenes were compared. The results of the K–W one-way ANOVA were RMS *χ*^2^(4) = 6.295 (*p* = 0.178) and MAV *χ*^2^(4) = 7.124 (*p* = 0.130), and the EMG time domain characteristics in the different scenes did not significantly differ: MPF *χ*^2^(4) = 14.410 (*p* = 0.006) and MF *χ*^2^(4) = 13.823 (*p* = 0.008). The EMG frequency domain characteristics were significantly different between the different scene experiences, indicating different degrees of arm muscle activity and body involvement in the different scenes, which supports the results that there were differences in the sense of presence in the different scenes.

After the EEG signal was filtered using AcqKnowledge 5.0, the artifacts were removed manually in order to prevent interference from events such as eye movements, large movements of the head and body, and sweating. Then, eye movement artifacts were detected using a sliding window function peak–peak threshold method with MATLAB, and amplitude changes of more than 150 μV were excluded. Drift and other artifacts larger than 100 μV were detected and marked by a cyclic algorithm, then excluded. In order to gather the power spectral density, the Welch method was used to divide the data into 1 s long windows with 50% overlap. The EEG indices of each channel were calculated, and the PSD was divided into α (8–13 Hz), β (13–30 Hz), β_low_ (13–15 Hz), β_high_ (23–30 Hz), and θ (4–8 Hz). EEG waveforms for each frequency band are shown in Figure 17. Then, three EEG indices were calculated: BBR = β_high_/β_low_, EI = β/(θ + α), and TBR = θ/β, representing alertness, engagement, and calmness, respectively [111].

After processing the missing data, the normal test showed that the data did not conform to a normal distribution, such that the use of test methods for the scene before the experience and the experience in the process of brain electrical characteristics, respectively, were through the Wilcoxon signed rank test, with the results for alertness (Z = 4.131, *p* < 0.001), participation (Z = 5.601, *p* < 0.001), and calmness (Z = 5.713, *p* < 0.001) indicating that there were significant differences in the EEG index before and during the experience. The measured differences in the power spectral density ratio for the EEG signals before and after the five scenarios are shown in Figure 18. Combined with the descriptive statistical results, prefrontal alertness and engagement were significantly increased and the calming signal index was decreased in the VR restorative environment experience, indicating that the prefrontal lobe was activated and in an excited state compared to the baseline state.

Comparing the EEG indicators when experiencing different scenes, the results of the K–W one-way ANOVA were as follows: engagement, *χ*^2^(4) = 0.808 (*p* = 0.937); calmness, *χ*^2^(4) = 3.116 (*p* = 0.539). The EEG signals of the different scenes had no significant differences in terms of engagement and calmness. For alertness (*χ*^2^(4) = 11.650, *p* = 0.020), the prefrontal EEG alert state index was significantly different in the different scenes, indicating that there were differences in the prefrontal lobe activity in different scenes. The results of the EEG signal analysis showed that, during the VR restorative environment experience, the alert state and engagement of the prefrontal lobe were significantly increased, while the signal in the calm state was weakened, indicating that, compared to the baseline state, the scene experience brought the prefrontal lobe into an excited state, which indeed contributed to individual cognitive recovery and improved cognitive function.

## 4. Discussion

### 4.1. Healing Effect of a VR Restorative Environment

In Study I, we proved that the VR restorative garden scene had a healing effect, consistent with previous research results. Compared to the physical environment, the VR forest environment had the same healing effect and was more attractive and coherent [49]; however, as the participants only experienced a visual scene with VR equipment (i.e., without other sensory experiences), the advantages of rich multi-sensory VR technology experiences, in terms of promoting emotional and behavioral responses, have not yet been fully exploited [112,113]. Therefore, by adding other sensory stimuli in Study II, the participants could temporarily get away from reality and forget their worries, through experiencing environmental content and interactive activities, thus enhancing their immersion, relaxation, and emotional relief. In addition, most participants were first-time users of VR devices and were not comfortable with the weight of the headset, handle operation, or vertigo caused by the brain’s asynchronous perception of movement [114,115], which affected the overall experience, to a certain extent; in future research, this should be taken into consideration, and timely attention should be paid to the physical and mental state of the participants. In addition, we provided participants with a quiet urban environment without people and discovered the healing effect of VR urban scenes. Previous studies have shown that stressed individuals prefer to be alone or with only a few persons to recover, a combination of refuge, nature and rich in species, and a low or no presence of social, which could be interpreted as the most restorative environment for stressed individuals [116], so the selection of the restorative environment may be not only different between the natural and urban environments, but may also be related to an individual’s environmental preferences and the physical properties of the scene itself, such as existence of people in the scene, its brightness, and color saturation.

### 4.2. Subjective Healing Due to Different Scene Experiences

In Study II, participants with mild-to-moderate anxiety and depression mood states were intervened. It was found that there were no significant differences between the different VR restorative environment scenarios in terms of individual subjective healing feeling. This may have been caused by the inter-subject design, which made individuals insensitive to the changes in scene reality and psychological immersion, such that the differences in the perceived restorative resilience of the scenes was not obvious. In addition, it is unknown whether the absence of sound in the VR scenes affected the associated environmental healing. Previous studies have confirmed that natural sound has a strong impact on recovery [117,118]. Sound is an important element to improve the immersion of virtual reality. The absence of sound further reduces the differences in the healing effects of the different scenes.

### 4.3. Differences in the Presence Experienced in Different Scenes

There were significant differences in the presence experienced in the different VR scenes, which may have been caused by motion sickness, device perception, personality traits, interaction with VR scenes, immersion, and participation [119,120,121,122,123]. EMG data feedback supported the differences in the sense of control and participation in the different scenes. We only found a difference in the sense of presence between the VR restorative environment interactive tour group and the other four scenes, which may be due to the fact that this scene’s experience mode broke the visual experience of the VR urban environment and this visual experience of the VR restorative environment without any operations. Autonomous handle control was added to the scene experience mode consisting of only visual experience, which allowed participants a certain sense of control; however, it differed from the more interesting interactive tasks in the VR restorative environment, such as interactive sightseeing, fishing, and watering activities. Accordingly, the participants quickly completed the exploration independently, and the limited sense of control and richness of scene content did not meet their expectations, which affected their sense of presence.

### 4.4. Mediating Effect of Presence

As the results showed, the sense of presence had different mediating effects on different dependent variables. When the independent variable was the score of RES, the mediating effect of presence on negative emotions was the strongest, followed by self-efficacy and positive emotions. Based on these findings, this study proved that the recovery impact of VR rehabilitative environment on people was probably realized through the presence of VR scenes. However, the mechanism of presence has not yet probed in depth in this research, and the impact related to the sense of presence have not fully found and controlled, so its mediating role in the effects of RES on emotions and self-efficacy requires further study.

### 4.5. Impact of a VR Restorative Environment Experience on Individual Emotions and Self-Efficacy

From the results, we determined the effects of the VR restorative environment experience on the positive and negative emotions and self-efficacy of individuals with mild-to-moderate anxiety and depression and found different directions of influence, which preliminarily proved the intervention effect of VR scene experiences on individual emotions and self-efficacy. Previous studies have found that groups with high-anxiety traits have a strong attention bias toward negative information, indicating that they are sensitive to negative information and find it difficult to reject a bad state after locking threats. They are more alert to negative stimuli and pay more attention to their internal world for a longer time, while non-high-anxiety groups tend to pay more attention to happy information. Their lockdown time is also longer [30,124,125,126]; however, this particular attentional bias may be enhanced in virtual reality, which may stimulate negative feelings, allowing participants to make different evaluations of the VR restorative environment and different interaction activities, which have different effects on emotional interventions.

When we consider the comparison between different scenes. We did not find differences in individuals with mild-to-moderate anxiety and depression, as we had assumed. However, according to the hypothesis, for those in the visual experience group, who did not need to learn and operate the controller, the task was less difficult than for the interactive experiencing group, and so, their participation and enjoyment were lower than in those who performed the fishing and watering activities; therefore, these factors influence the presence due to VR [127]. Some studies have shown that emotions in the virtual environment are correlated with a sense of presence [128], which also explains the interaction between presence and emotion [30]. Therefore, we believe that such an association is the reason why individuals achieved better positive emotion enhancement in these scenes. However, due to the inherent defects of the design between groups, individuals were not allowed to experience all five different scenes, thus making the sensitivity of scene changes weak and leading to no significant differences in the healing effect. In addition, participants had expectations for the VR scene experience and, when the impact of scene differences on emotion and self-efficacy was less than their expectations, there was no difference between the groups across the different scene experiences. The results of this study are consistent with the results of previous research on VR restorative environments, where the healing effect of the environment may be far greater than the impact of media or activity mode changes [49]. In future research, if the within-group design with more sensitivity to experimental processing is adopted to increase individuals’ perceptions of differences in different scenes, other experimental results may be obtained.

We found that VR visual experiences in urban environments significantly reduced negative emotions, but did not have significant effects on individual positive emotions or self-efficacy. The reason for this discrepancy was that most of the participants recruited for the experiment were students who had been studying or working indoors for a long time. The VR urban scene was different from their pressure environment familiar to the participants, thus meeting one of the characteristics of a restorative environment: the feeling of being far away. Empty urban environments, which are free from crowding and oppression, and experiences without directed attention have a restorative effect [89]. Therefore, from another perspective, we also found that not only the natural environment, but also the urban scene, may have healing effects on those people who have been indoors and in stressful environments for a long time.

### 4.6. Effect of a VR Restorative Environment on Cognitive Function

Three brain function indicators were selected in this study, namely, alertness, engagement, and calmness. Through EEG feedback, differences in the effects of the five experimental scenes on the prefrontal lobe were found, which preliminarily verified the correlation between the VR restorative environment and individual cognitive function recovery. However, the mechanism of its influence is not clear yet. In addition, the EMG results showed that the scene experience enhanced arm muscle contractive activity, reflecting increased body involvement, and that the different interaction groups did not produce cognitive burden.

### 4.7. Innovation and Significance

In general, this study proves that the experience of a VR-based restorative environment scene has a good healing effect in people with mild-to-moderate anxiety and depression, where such recovery was reflected in its influence on the positive and negative emotions, self-efficacy, and cognitive function of these individuals. The influencing factors and the internal psychological mechanism of the recovery effect of restorative environments have been widely treated as the research focus of environmental psychologists; however, previous research on restorative environments has mostly been based on real scenes or non-interactive immersive VR scenes. In this study, a restorative environment was achieved by VR technology and 3D dynamic interactive activities, thus enhancing the connection between environment and people, which was lacking in previous studies. The interactive tasks focused on the activities of people in the environment, which enhanced their interests and avoided the adverse emotional reactions caused by a lack of interaction for a long time and prevented its effects on physiological indicators. On the basis of this innovation, research on the connection between the environment, behavior, and psychological experience can be supplemented to explore the differences in healing and the presence of experience due to different scenes, as well as the influence of restorative environments on individual emotions in VR scenes. We also used EEG feedback to prove the effect of the VR restorative environment on cognitive recovery, serving as a useful supplement to the research of restorative environments on individual cognitive recovery.

In theory, this study was a theoretical exploration, and the extension of research on the restorative environment will help to improve the understanding of the internal mechanism of the recovery due to the restorative environment. At the same time, it provides a theoretical basis for the practical application of environmental recovery. VR technology has a good restorative function and application value when people cannot access natural environments with high recovery potential [129]. This technology makes it possible for urban office workers, students, and special groups (e.g., individuals with disabilities) to effectively recover in a restorative environment. Therefore, this study has both theoretical value and practical significance. Although VR technology has drawbacks, such as high cost, difficulty of replacing real natural environments, and the ethical risk of accelerating separation from the natural world, considering the high application value of this technology and the trend of continuous development and improvement, we can look forward to an era in which VR serves millions of families.

### 4.8. Limitations

In this study, we proved that VR restorative scenes have a certain healing effect on mild-to-moderate anxiety and depression, but have the following shortcomings. First of all, considering the integration of psychological and VR technology, the attention, focus, and cognitive preference in the psychosocial response due to specific anxiety and depression moods have not yet been fully grasped. We failed to discuss the possible effects of attention [130], environmental preference [131], familiarity with the environment, and different types of recreation environments [132] on recovery outcomes.

Second, natural sound is one of the factors affecting the efficacy of restorative environments [133], as the perception of natural sound aids in mental recovery [134]. In this study, we only discussed the effects of restorative environments from the perspective of vision, and did not include auditory stimuli, which may have a greater impact on healing. In future research, sound can be added to the experience of VR scenes in order to explore the recovery effect of restorative environments in a more comprehensive way. In addition, it was found that some participants in this study began to become a little upset after 5 min of experiencing the scene due to completion of the required activities, or familiarity with the environment, or as they tried to explore whether there were any further tasks that needed to be completed in the VR environment. In future research, the balance between scene design and time setting can be further considered, and the interference of irrelevant variables can be controlled by increasing the richness of the experimental scenes and the complexity of interaction activities, or by shortening the experience time. In addition, we found that the occurrence of motion sickness and physical discomfort due to the immersive VR scene experience could affect the emotional state of participants and the pleasure of the experience, to some extent [30]. Finally, different VR scene experiences will bring different senses of presence, which may be one of the reasons for the differences observed in the healing effect.

Third, the improvement of anxiety and depression is a long-term endeavor. In this study, we tested the participants only once. In next research, we can discuss whether this kind of intervention can benefit the individuals with mild-to-moderate anxiety and depression in a long-term way.

In future research, we may explore the regulating effect of the sense of presence on the mental and physical healing effects due to VR restorative environments, as well as further explain the associated internal psychological mechanism. In summary, future research on restorative environments based on the assumption that researchers can comprehensively consider the psychological needs, physiological responses, cognitive patterns, operating habits, and environmental preferences of users in the VR environments, as well as discuss how to improve the recovery effect of VR environments and apply it in practice.

## 5. Conclusions

Considering the results of this study, we drew the following conclusions:There was no significant difference in the healing effect between different VR scenes, but the restorative score of the VR urban scene was higher than that of the VR natural environment.A high sense of presence could be experienced in different VR scenes, and interactive activities in VR scenes can provide a great presence experience. However, roaming in a natural environment through controller operation had the lowest sense of presence. The differences of presence were also reflected in EMG.The recovery effects of VR restorative environment on emotion and self-efficacy are realized through the presence of VR scenes.VR restorative environments are helpful for emotional improvement and cognitive recovery in individuals with mild-to-moderate anxiety and depression. VR urban scenes also have good recovery effects. In terms of cognitive recovery, self-efficacy improved significantly. In addition, from the perspective of EEG indicators, the VR restorative scene experience activated the prefrontal lobe, which is conducive to cognitive recovery in individuals with mild-to-moderate anxiety and depression. In terms of emotional improvement, negative emotions were significantly reduced in the different VR scene groups.

## Figures and Tables

**Figure 1 ijerph-18-09053-f001:**
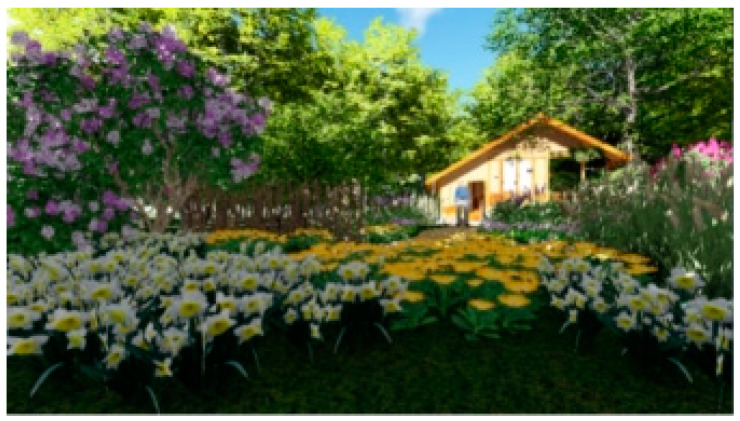
Example 1 of an experimental picture.

**Figure 2 ijerph-18-09053-f002:**
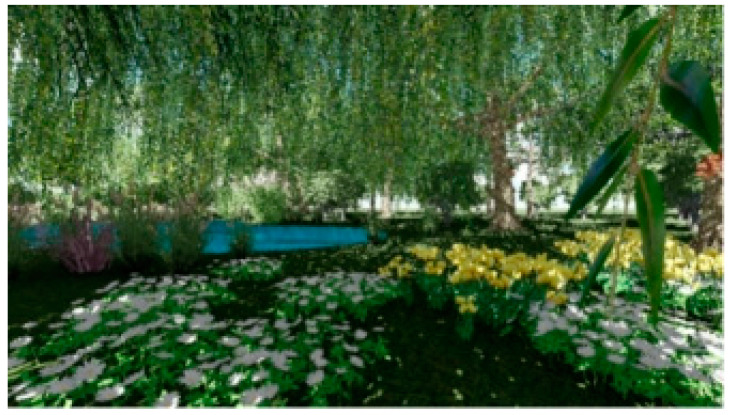
Example 2 of an experimental picture.

**Figure 3 ijerph-18-09053-f003:**
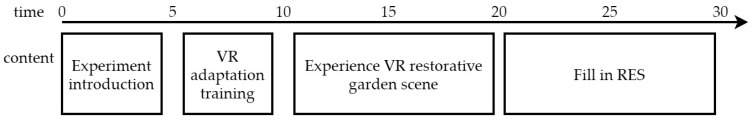
Experimental procedure. VR: Virtual Reality, RES: Restorative Environmental Scale.

**Figure 4 ijerph-18-09053-f004:**
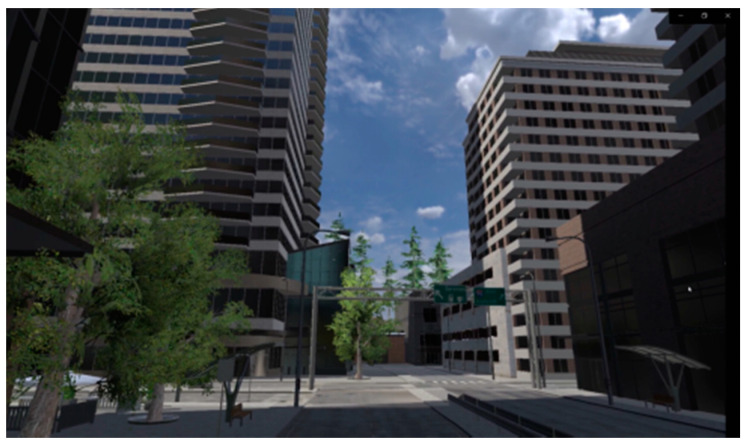
Screenshot 1 of an urban scene.

**Figure 5 ijerph-18-09053-f005:**
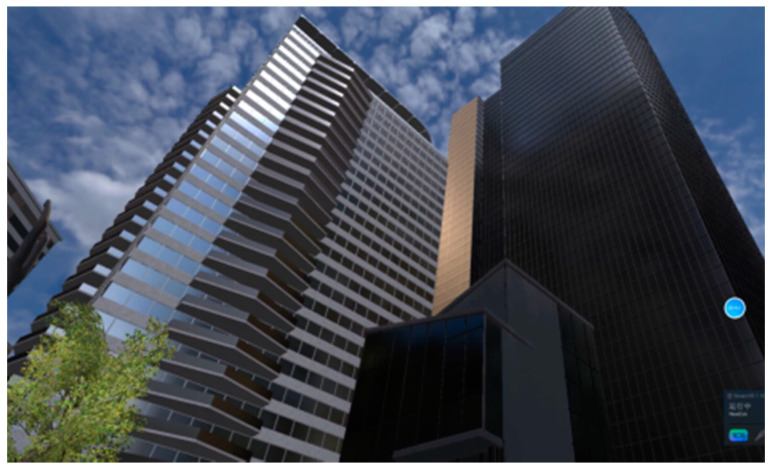
Screenshot 2 of an urban scene.

**Figure 6 ijerph-18-09053-f006:**
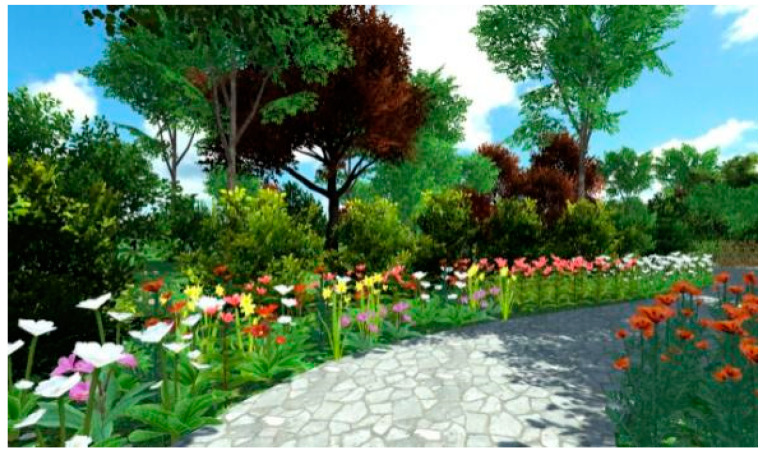
Screenshot 1 of a restorative garden scene.

**Figure 7 ijerph-18-09053-f007:**
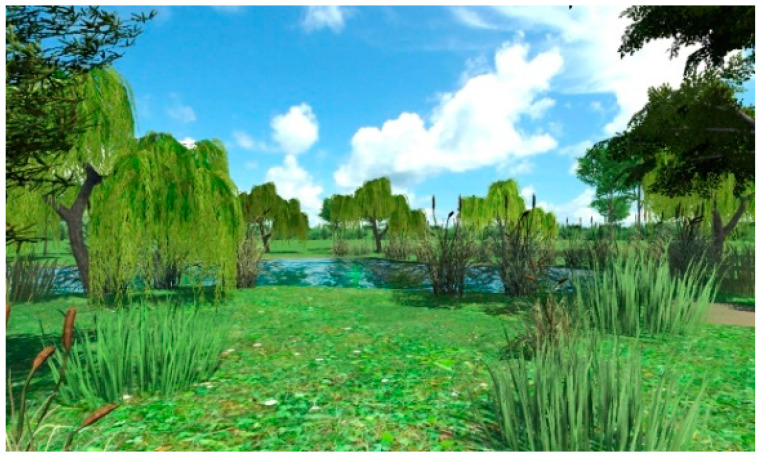
Screenshot 2 of a restorative garden scene.

**Figure 8 ijerph-18-09053-f008:**
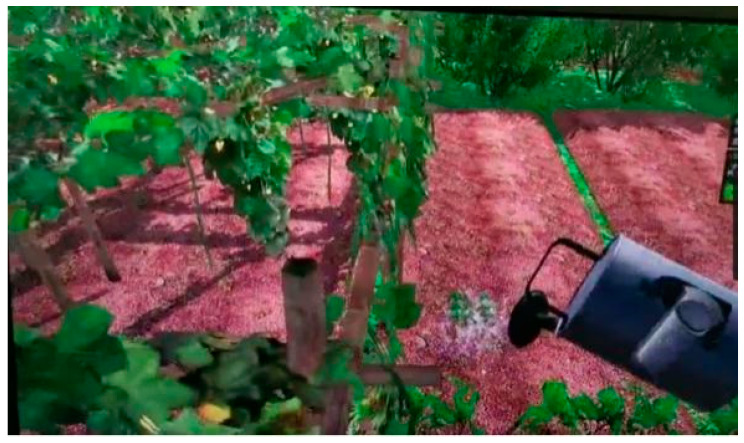
Screenshot of the interactive activity in a restorative garden: watering.

**Figure 9 ijerph-18-09053-f009:**
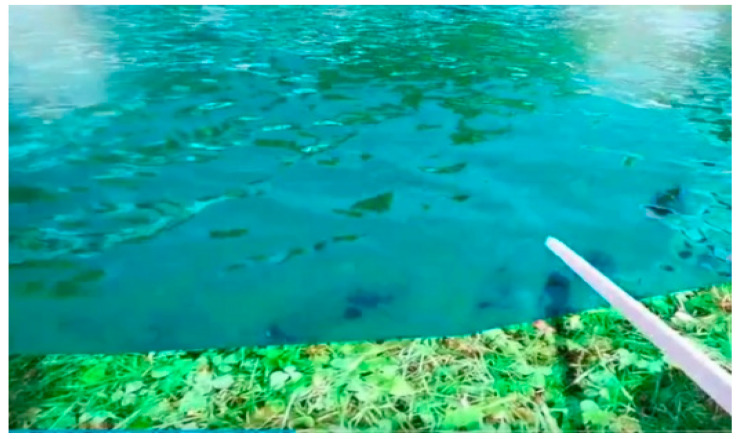
Screenshot of the interactive activity in a restorative garden: fishing.

**Figure 10 ijerph-18-09053-f010:**
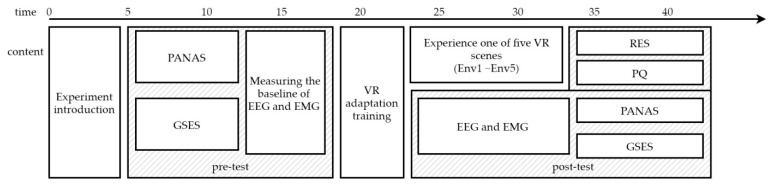
Experimental procedure. PANAS: Positive and Negative Affect Scale, GSES: General Self-Efficacy Scale, EEG: Electroencephalography, EMG: Electromyography, PQ: Presence Questionnaire.

**Figure 11 ijerph-18-09053-f011:**
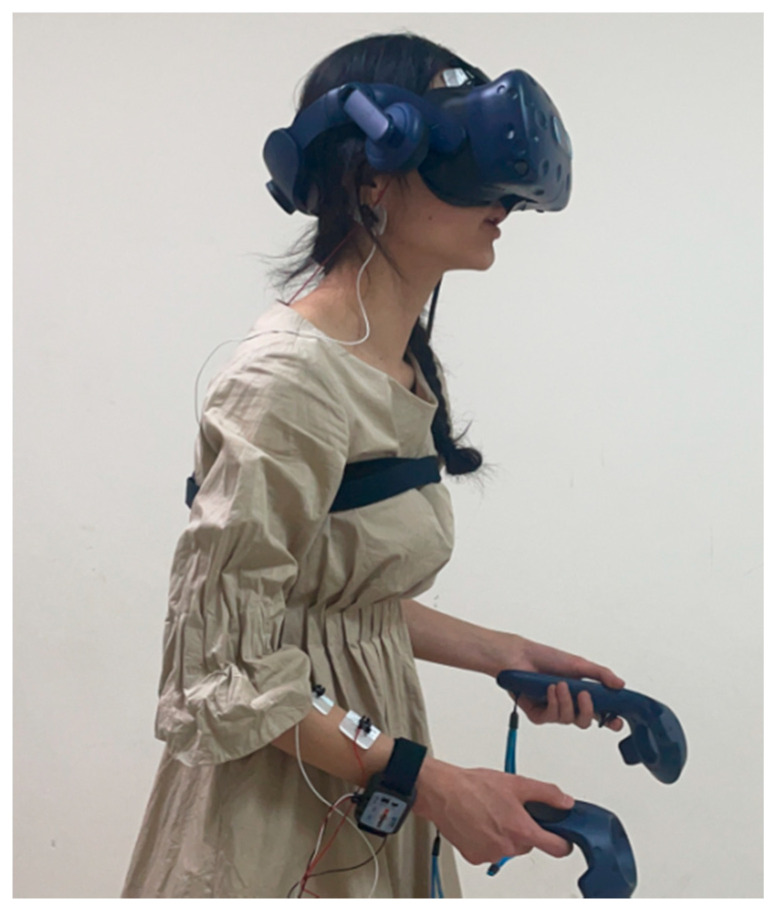
Experiment scene schematic 1.

**Figure 12 ijerph-18-09053-f012:**
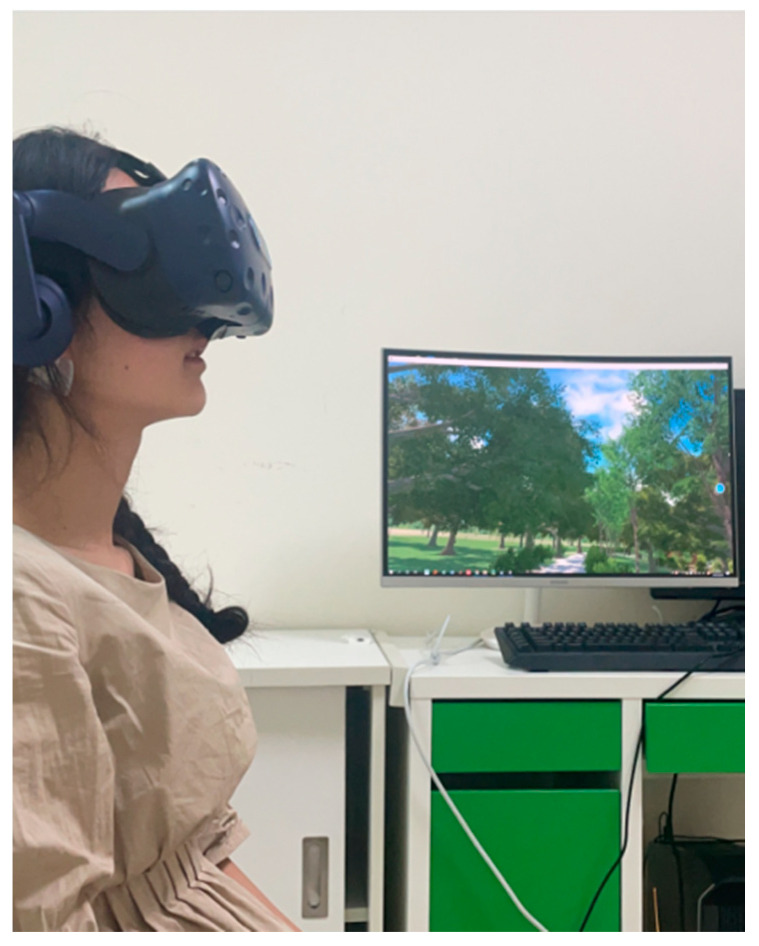
Experiment scene schematic 2.

**Figure 13 ijerph-18-09053-f013:**
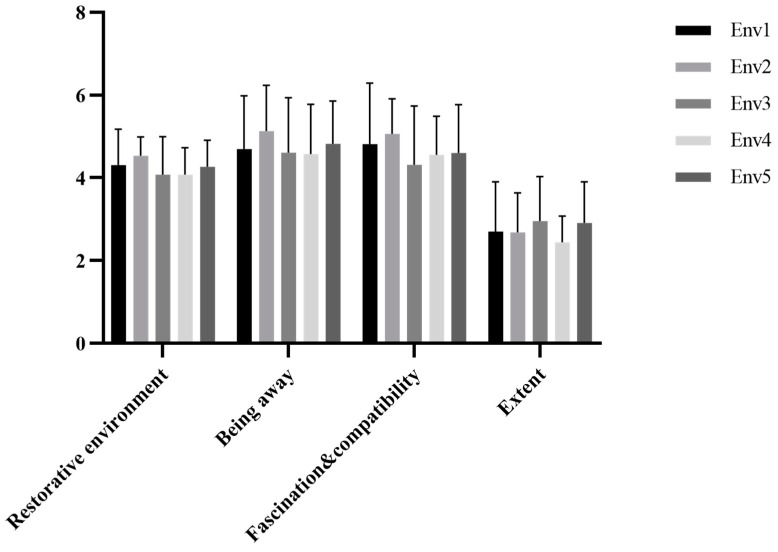
Descriptive statistical results of the restorative environment and each dimension. Interpretation: Env1, VR urban environment visual experiencing group; Env2, VR restorative environment visual experiencing group; Env3, VR restorative environment interactive experiencing group; Env4, VR restorative environment with fishing interaction group; Env5, VR restorative environment with watering interaction group.

**Figure 14 ijerph-18-09053-f014:**
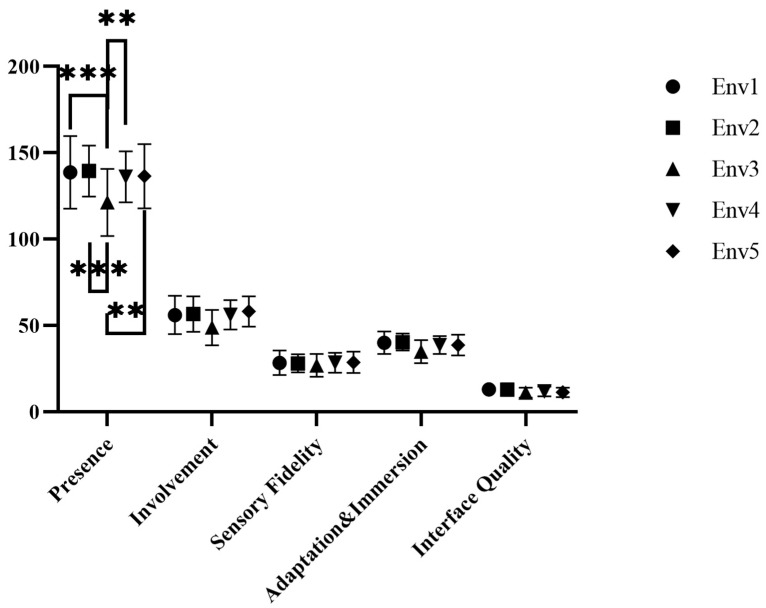
The total score of presence and each dimension score. Interpretation: ** *p* < 0.01, and *** *p* < 0.001. Env1, VR urban environment visual experiencing group; Env2, VR restorative environment visual experiencing group; Env3, VR restorative environment interactive experiencing group; Env4, VR restorative environment with fishing interaction group; Env5, VR restorative environment with watering interaction group.

**Figure 15 ijerph-18-09053-f015:**
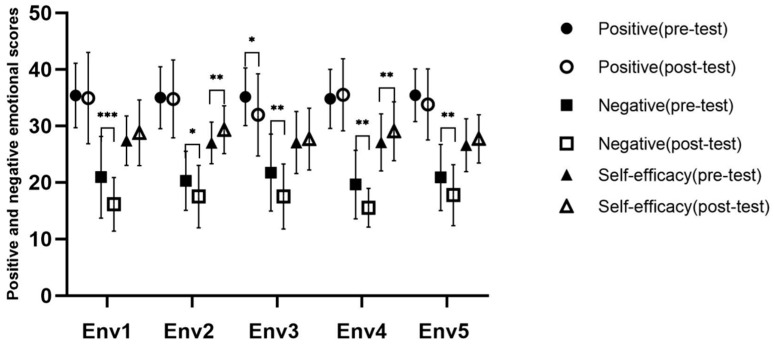
The score of positive and negative emotions and self-efficacy score. Interpretation: * *p* < 0.05, ** *p* < 0.01, and *** *p* < 0.001. Env1, VR urban environment visual experiencing group; Env2, VR restorative environment visual experiencing group; Env3, VR restorative environment interactive experiencing group; Env4, VR restorative environment with fishing interaction group; Env5, VR restorative environment with watering interaction group.

**Figure 16 ijerph-18-09053-f016:**
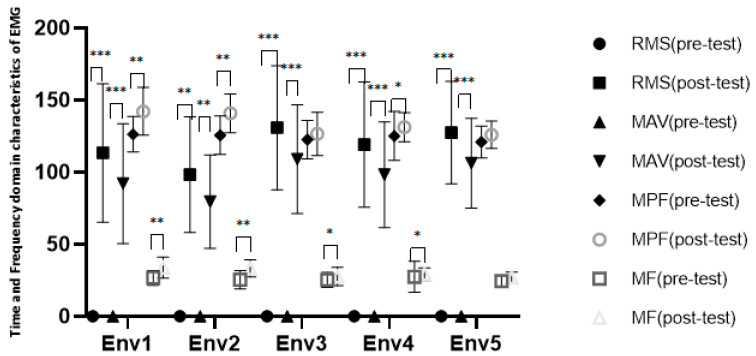
Time and frequency domain characteristics of EMG. Interpretation: * *p* < 0.05, ** *p* < 0.01, and *** *p* < 0.001. Env1, VR urban environment visual experiencing group; Env2, VR restorative environment visual experiencing group; Env3, VR restorative environment interactive experiencing group; Env4, VR restorative environment with fishing interaction group; Env5, VR restorative environment with watering interaction group.

**Figure 17 ijerph-18-09053-f017:**
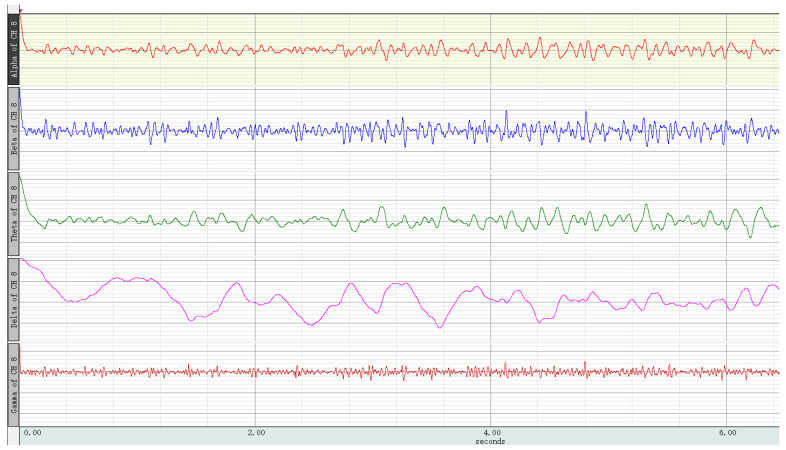
Example of EEG waveforms at various frequency bands.

**Figure 18 ijerph-18-09053-f018:**
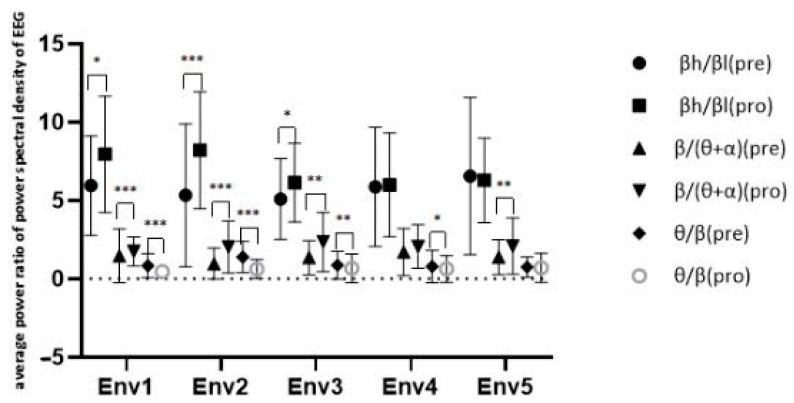
Average power ratio of the power spectral density of EEG. Interpretation: * *p* < 0.05, ** *p* < 0.01, and *** *p* < 0.001. Env1, VR urban environment visual experiencing group; Env2, VR restorative environment visual experiencing group; Env3, VR restorative environment interactive experiencing group; Env4, VR restorative environment with fishing interaction group; Env5, VR restorative environment with watering interaction group.

**Table 1 ijerph-18-09053-t001:** Analysis of the mediating effect of presence.

Dependent Variables		Effect	Boot SE	Boot LLCT	Boot ULCI	Relative Effect
Positive	Total Effect	0.1045	0.0241	0.0000	0.0569	
	Direct Effect of Presence	0.0604	0.0260	0.0217	0.0090	57.80%
	Mediating Effect of Presence	0.0441	0.0150	0.0169	0.0748	42.20%
Negative	Total Effect	−0.0124	0.0318	0.6966	−0.0753	
	Direct Effect of Presence	0.0325	0.0350	0.3554	−0.0368	41.99%
	Mediating Effect of Presence	−0.0449	0.0200	−0.0881	−0.0108	−58.01%
Self-efficacy	Total Effect	0.0485	0.0216	0.0263	0.0058	
	Direct Effect of Presence	0.0208	0.0239	0.3855	−0.0265	42.89%
	Mediating Effect of Presence	0.0277	0.0154	0.0004	0.0611	57.11%

Interpretation: Boot SE, Boot LLCI, and Boot ULCI, respectively, refer to the standard error of the indirect effect estimated by the deviation-corrected percentile Bootstrap method, and the lower and upper limits of the 95% confidence interval.

**Table 2 ijerph-18-09053-t002:** Paired sample *t*-test results of positive and negative affect and general self-efficacy.

Environment	Pre–Post	df	*t*	*p*
Env1	positive0–positive1	28	0.657	0.517
	negative0–negative1		4.633 ***	0.000
	self-efficacy0–self-efficacy1		−2.036	0.051
Env2	positive0–positive1	23	0.207	0.838
	negative0–negative1		2.802 *	0.010
	self-efficacy0–self-efficacy1		−3.268 **	0.003
Env3	positive0–positive1	29	2.315 *	0.028
	negative0–negative1		3.397 **	0.002
	self-efficacy0–self-efficacy1		−0.975	0.337
Env4	positive0–positive1	30	1.707	0.098
	negative0–negative1		3.770 **	0.001
	self-efficacy0–self-efficacy1		−3.875 **	0.001
Env5	positive0–positive1	31	1.707	0.098
	negative0–negative1		3.770 **	0.001
	self-efficacy0–self-efficacy1		−2.112	0.043

Interpretation: * *p* < 0.05, ** *p* < 0.01, and *** *p* < 0.001.

## Data Availability

No publicly archived datasets analyzed or generated were used in this study.

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
