# Peer review of "Effect of a Virtual Reality-Based Restorative Environment on the Emotional and Cognitive Recovery of Individuals with Mild-to-Moderate Anxiety and Depression"

_ijerph, 2021, doi:10.3390/ijerph18179053_

Round 1

Reviewer 1 Report

This paper focuses on building a restorative environment in virtual reality for the study of emotional disorders, and this work is very valuable. However, I have some comments and suggestions which hopefully would help perfecting the manuscript. Main question is that there is no obvious difference between the construction of restorative environment scene and other ordinary virtual reality scenes. My comments and suggestions are as follows.

  1. As we know, researches on anxiety and depression recovery in virtual reality are very common. It seems to be no obvious difference between the restorative environment scene proposed in this paper and other research scenes. So I hope the author should emphasize the difference between restorative environment and other virtual reality scenes.
  2. There are some relevant studies have proved that using virtual scene for education can improve people's emotion and consciousness. (e.g. , “An augmented represen- tation method of debris flow scenes to improve public perception”,“We cannot let this happen again”: reversing UK floodpolicy in response to the Somerset Levels floods ”,etc). So the reviewer suggests the author should further emphasize the role and significance of virtual scene in the introduction.

  1. " Based on environmental psychology…. which included seven environmental zones: Forest area, meditation area, aroma area, lawn area, gardening area, rest and interaction area, and water scenic area. " . This sentence has mentioned seven environmental zones, including aroma area. However, the construction and function of aroma region have not been proposed in the paper, and the experimental results of aroma region have not been mentioned. So I hope the authors should explain it in detailed.
  2. In this paper, there is no sound element in the virtual reality scene, but sound is the basic element to improve the immersion of virtual reality. Whether this will lead to inaccurate experimental results or unsatisfied effects. I hope the author explain it briefly.
  3. Page14, the interaction of watering and fishing in this experimental design is too simple, I'm confused that whether these simple interactive behaviors will lead to the differences in the experimental results. So I hope the author should explain it in detailed.
  4. The improvement of anxiety and depression is a long-term work. The participants in this paper only conducted one test. I think it is difficult to explain the improvement of anxiety and depression in a restorative environment. So I hope the author should explain it in detailed.

Reviewer 2 Report

Brief summary

This manuscript presents a main experiment (N=195) and two pre-studies (N=8 and N=26), exploring the effect of virtual natural environments vs. a virtual urban environment on emotional and cognitive recovery. The authors refer to recognized recovery theories (SRT and ART) which deal with emotional and cognitive recovery.

In the main experiment, the effects of different VR restorative environments on the emotional and cognitive recovery of individuals with mild to moderate anxiety and depression were tested. Moreover, the sense of presence in VR scenes and EMG and EEG data were performed.

The manuscript provides three central hypotheses guiding the work and tests these using a series of (paired) t-tests, one-way ANOVAS and post-hoc tests. The results showed that the restorative environment images (pre-test) and the VR scenes had a recovery effect (reducing negative emotions and increasing self-efficacy). However, there was no difference in the subjective feeling of recovery among the different scenes. Contrary to expectation, the recovery score of the VR urban environment (= control group) was higher than that of the natural environment. The study found a high sense of presence in different VR scenes. Moreover, the VR restorative scenes activated the prefrontal lobe, which is beneficial to the cognitive recovery.

This research gives contributing insights into a natural VR intervention for individuals with mild to moderate anxiety and depression.

For the authors:

I appreciate the effort that has gone into this work and thank you for your consideration in putting together this manuscript. It is evident that you have considered the applied value of this work by extending pre-tests and a huge main study and in selecting the special sample of individuals with mild to moderate anxiety and depression who are most likely to benefit from this intervention. It seems to be capitalizing on technological advances and what these might mean for future therapy interventions.

Originality/Novelty: Is the question original and well defined? Do the results provide an advance in current knowledge?

The research question of this study was to explore the intervention effect of the presence of a VR restorative environment on individuals with mild-to-moderate anxiety and depression from the perspective of emotion and cognition.

The derivation of the hypotheses does not emerge completely from the theory section. The idea to test participants with mild-to-moderate anxiety and depression is ‘original’. However, the authors do not explain in their theoretical background why they focus on this special sample - why should these people particularly benefit from the intervention compared to healthy people? And why were the pre-tests still done with healthy individuals?

The theoretical background includes the common recovery theories (ART & SRT), which have been predominantly tested on healthy subjects in former studies. Thus, it would be helpful to add a separate paragraph justifying the reference to the special sample (individuals with low-to-moderate anxiety and depression).

Further, the authors do not yet clearly elaborate the research gap and former research in their theory part. For example, there are former VR-studies investigating the effect of a Virtual Reality-Based Restorative Environment on the Emotional Recovery of Individuals that are not mentioned in the current work (e.g., Karacan et al., 2020; Schutte et al., 2017; see also the current review from Mollazadeh & Zhu, 2021 about ‘Application of Virtual Environments for Biophilic Design: A Critical Review’). It would be nice if the following question could be answered: What exactly is the added value of the present study compared to those already published?

The authors argue that it is new and contrary to expectations that the VR urban environment was perceived more restorative than the VR natural environment. This idea is already published for real environments (instead of VR), so please add the theoretical derivation. The authors presented a quiet urban environment without people. For instance, Grahn and Stigsdotter (2010) pointed out that stressed individuals prefer to be alone or with only a few persons to recover; see also the concepts of territoriality, privacy, and autonomy (Richter, 2008).

The authors name three main findings in their abstract:

1) Both the restorative environment images and the VR scenes had a healing effect - I would recommend to introduce a definition what the term ‘healing effect’ exactly means.

2) Another finding is that a high sense of presence can be experienced in different VR scenes. However, this is not a new insight, there are several former studies showing this fact.

3) A VR restorative environment is helpful for the emotional improvement and cognitive recovery of individuals with mild-to-moderate anxiety and depression. Indeed, investigating individuals with mild-to-moderate anxiety and depression in VR is a new topic. However, I miss a consistent approach in the study selection and verification of the VR natural scenes: Why have you done the pre-test with healthy participants (N=26) and later, you test participants (N=195) with mild-to-moderate anxiety and depression?

Significance: Are the results interpreted appropriately? Are they significant? Are all conclusions justified and supported by the results? Are hypotheses and speculations carefully identified as such?

The authors report for Experiment 1 (Recovery Validation of Scenes) and Experiment 2 (Verification of Recovery in a VR Environment) as well as Study II (Presence and Recovery Effect of Experiencing a VR Restorative Environment) that the “RES were found to be normally distributed”. I would like to ask for the respective statistics.

For Experiment 1 and Experiment 2, there are no formulated hypotheses, therefore it is actually not possible to assess the result parts. Apart from that I found it surprising that the authors compared the RES values with the mean of the RES scale in Experiment 2 – as I said, the derivation of the hypotheses is missing, so it would be very helpful to add the corresponding hypothesis and the derivation.

Moreover, the authors mention that they found significant differences, but they are not reporting the direction of difference, for instance:

P 7, L 266: “The results showed that there was no significant difference between the total score and the mean of the scale (t(15) = 0.696, p > 0.05), but the three dimensions (tbeing away (15) = 2.335, p < 0.05; tattraction and compatibility (15) = 2.668. p < 0.05; trich (15) = −3.996, p < 0.01) were significantly different from the average score (4).”

P 5, L 210: I found it somewhat surprising that the subscales fascination and compatibility are considered as one factor in the Chinese version of RES. In contrast, the Perceived restorativeness scale (PRS) is postulating two separate factors for fascination and compatibility. Thus, I would like to see the factor analysis of the RES questionnaire.

Unfortunately, I could not find the publication of the RES scale:

P 23, L 881: Ye, L.H.; Wu, J.P. Development of the Recovery Environment Scale. Chin. J. Health Psychol. 2010, 18, 1515–1518.

Study II (main study) was performed to test three hypotheses:

P 7, L 289-290:

H1: “There will be differences in subjective restoration and the sense of presence of different VR restorative scenes.”

I find it somewhat vague to postulate just “differences in subjective restoration and the sense of presence” without any idea which of the presented intervention groups could be better or worse and why – I lack the reference to the theoretical foundation. Moreover, in the method section you are writing that the VR urban environment was added as a control group. Thus, why are no differences expected to the control group?

P 11, L 407 - 426: The results of the first part of H1 are described in section “3.4.1. Differences in the Environmental Recovery of Different Scene Experiences”. There are some “small errors” in reporting regarding the following statement:

P 12, L 423: “Therefore, it was found that the selection of the healing environment was not only different between the natural and urban environments, but may also be related to an individual's environmental preferences and the physical properties of the scene itself, such as its brightness and color saturation.”

The differences just emerge for descriptive statistics, not for inferential statistics. Thus, there is no statistical difference or a benefit for the urban environment in comparison to the natural ones - that would be a wrong conclusion.

P 12, L 424: Moreover, the sentence “but may also be related to an individual's environmental preferences and the physical properties of the scene itself” is a personal interpretation/idea. Thus, it shouldn’t be presented in the results section.

P 12, L 426: Further, the authors mention that the brightness and color saturation was not constant in all intervention groups. This is a problem, because then you have several independent variables and you cannot say clearly what you are actually testing (differences in color, saturation, VR nature scene). The objectivity is missing here. Please manipulate just one variable and try to (statistically) control all other parameters.

P 12, L 433 – 444: The results of the second part of H1 are described in section “3.4.2. Differences in the Presence Experienced in Different Scenes”. Again, the authors report that data were normally distributed without reporting the statistics. Please, also report statistics.

P 12, L 349 – 440: Results showed that “there were significant differences in the sense of presence for the different scenes.” Thus, I would expect that in further analysis, the sense of presence will be involved in the analyses (as a moderator) to allow comparisons between groups. However, this is not the case.

P 7, L 291 – 292:

H2: “VR restorative scenes will contribute to the mood improvement and cognitive recovery of individuals with mild-to-moderate anxiety and depression.”

P 13, L 451: The results of H2 are described in section “3.4.3. Effect of VR Restorative Environment Experience on Emotion and Self-Efficacy”.

P 13, L 452 - 453: The authors report that they have excluded “outliers from the positive and negative emotional and self-efficacy scores of the five experimental groups before and after experiencing VR scenes”. Please explain the reason for this decision. It's not possible to exclude participants just because they are outliers without giving a comprehensible reason.

P 13, L 455 – 457: “A paired samples t-test was conducted for the positive and negative emotional and self-efficacy scores of the five experimental groups before and after experiencing the VR scenes.” There could be an alpha error accumulation, since you performed paired t-tests with more than two conditions (in your case five conditions). Please perform a repeated measurement ANOVA.

P 14, L 488: In Table 1 the standard deviation is quite large, is this correct? You report “M”, but you mean the difference of the mean pre – mean post, right?

Moreover, it’s interesting that for Env 1, Env 2, Env 3 and Env 5 the positive effect was descriptive higher before intervention compared to after intervention. For Env 3, there is also a significant difference, showing that the positive effect was lower after intervention. Unfortunately, the authors interpret the effect in the wrong direction:

P 13, L 470:“In the Env3 condition, for positive and negative emotions, there were significant differences before and after the VR scene experience.[…]

P 13, L 473 - 475: “the interactive experience in the VR restorative environment significantly increased positive emotions and significantly reduced negative emotions..”

P 14, L 490: “All of the five virtual scenes significantly reduced negative emotions.”

I would expect that the control condition (=VR urban environment) should not decrease negative emotions, is it?

P 7, L 293 – 296:

H3: “Different VR restorative scenes will have different degrees of emotional and cognitive recovery for individuals with mild-to-moderate anxiety and depression: VR restorative scenes will help to improve self-efficacy, while the VR urban scenes will  aggravate negative emotions and reduce self-efficacy.”

The hypothesis is not sufficiently clear. Why have the authors not expected that VR restorative environments enhance positive emotions and decrease negative emotions (see SRT in the theory part)?

P 14, L 499 - 509: The results of H3 are described in section “3.4.4. Degree of Emotional and Self-Efficacy Recovery Due to Different VR Experiences”

In the section before, you reported “Effect of VR Restorative Environment Experience on Emotion and Self-Efficacy”, it seems very similar to me.?

P 14, L 500-501: The authors perform a difference score for pre- and post scores of positive and negative emotions and self-efficacy in each experimental group in order to obtain their changes. In former analyses (H2), the authors have tested mean differences pre/post. Is there a reason for this change in the analysis? I also miss a table summarizing descriptives and test results.

P 15, L 510: There is also a section called “3.4.5. EMG and EEG Feedback from Different Scene Experiences” in the results part, but no concrete hypothesis is formulated. I also miss the theoretical derivation. Later, in the discussion, the authors explain that VR restorative scene experiences “activated the prefrontal lobe, which is conducive to cognitive recovery in individuals with mild-to-moderate anxiety and depression”, but they are not explaining this circumstance in the theoretical background.

Quality of Presentation: Is the article written in an appropriate way? Are the data and analyses presented appropriately? Are the highest standards for presentation of the results used?

The article is written in an appropriate way. However, not all presented analyses fit the hypotheses made. Thus, the standards of the presentation of the results are not completely fulfilled, for instance in section 3.4.4. a table presenting the results is missing.

I also miss the link between different analyses: For example, in H1 the authors found differences in subjective restoration and the sense of presence. Thus, I would have expected that the authors would include the differences in subjective restoration and sense of presence in further analyses, in form of mediators. However, this was not the case.

Scientific Soundness: Is the study correctly designed and technically sound? Are the analyses performed with the highest technical standards? Are the data robust enough to draw the conclusions? Are the methods, tools, software, and reagents described with sufficient details to allow another researcher to reproduce the results?

P 4, L 186 - 196: The authors describe in section “2.1. Design and Implementation of a VR Restorative Environment” seven environmental zones: Forest area, meditation area, aroma area, lawn area, gardening area, rest and interaction area, and water scenic area. In the later course of the paper, no further reference is made to the individual zones and their specific effects. For example, which aromas were presented, how long, in which intensity, what aroma application (e.g. scent diffuser) was used?

P 5, L 197 - 202: In section “2.2. Experiment 1: Recovery Validation of Scenes” the authors have tried a validation of the recovery potential of the presented scenes. Unfortunately, just 10 participants were tested and two of them were excluded due to a normality check – I was wondering: What was the questions to decide that two participants are not ‘normally’? With such a small sample, a test of normality is not meaningful.

P 6, L 255-256: “the experimenter asked the participants about their feelings.” The interaction of experimenter and subject should be as low as possible to avoid favor hypothesis-compliant behavior (see Rosenthal effect). Was it not possible to ask the subjects in a standardized way, e.g., with an online questionnaire?

P 6, L 228: In the results section (2.2.4.) of Experiment 1 the authors conclude: “…the results indicate that the initial restorative scenes had a certain recovery effect, in which the characteristics of being away, attraction, and compatibility were better” - better than what?

P 6, L 236: In section “2.3. Experiment 2: Verification of Recovery in a VR Environment” the authors do not clearly state the used experimental design. Further, I assume that the four scenes were presented always in the same order to all participants. I would recommend a randomized presentation to avoid sequence effects and to compensate carry-over-effects.

P 7, L 272: Also in section “3. Study II: Presence and Recovery Effect of Experiencing a VR Restorative Environment” the authors does not name the experimental design (between vs. within subject).

Interest to the Readers: Are the conclusions interesting for the readership of the Journal? Will the paper attract a wide readership, or be of interest only to a limited number of people? (please see the Aims and Scope of the journal)

The present form of the manuscript has several methodological problems, e.g. problems with content objectivity. Therefore, at the current state of the article, I cannot yet assess if the conclusions are interesting for the readership of the Journal.

Overall Merit: Is there an overall benefit to publishing this work? Does the work provide an advance towards the current knowledge? Do the authors have addressed an important long-standing question with smart experiments?

There is an overall benefit to publishing a strongly revised version of this work. The work provides an advance towards the current knowledge as it examined a special sample, namely individuals with mild-to-moderate anxiety and depression. Unfortunately, the pre-testing of the selected VR scenes does not seem objective to me.

English Level: Is the English language appropriate and understandable?

The English level is appropriate and understandable. In short, I can see potential and opportunities for contribution in this work but think that there are some big issues to work through to get it there. I hope that my feedback can provide some helpful ideas for doing this.

Round 2

Reviewer 1 Report

All my concern have been revised, I do not have further comments.